# The Medium Cut-Off Membrane Does Not Lower Protein-Bound Uremic Toxins

**DOI:** 10.3390/toxins14110779

**Published:** 2022-11-10

**Authors:** Yang Gyun Kim, Sang Ho Lee, Su Woong Jung, Gun Tae Jung, Hyun Ji Lim, Kwang Pyo Kim, Young-Il Jo, KyuBok Jin, Ju Young Moon

**Affiliations:** 1Division of Nephrology, Department of Internal Medicine, Kyung Hee University School of Medicine, Seoul 05278, Korea; apple8840@hanmail.net (Y.G.K.); lshkidney@khu.ac.kr (S.H.L.); ha-ppy@hanmail.net (S.W.J.); 2Department of Biomedical Science and Technology, Kyung Hee Medical Science Research Institute, Kyung Hee University, Seoul 02453, Korea; jgt4047@gmail.com (G.T.J.); dlaguswl1001@khu.ac.kr (H.J.L.); kimkp@khu.ac.kr (K.P.K.); 3Department of Applied Chemistry, Institute of Natural Science, Global Center for Pharmaceutical Ingredient Materials, Kyung Hee University, Yongin 17104, Korea; 4Division of Nephrology, Department of Internal Medicine, Konkuk University Medical Center, Seoul 05029, Korea; nephjo@kuh.ac.kr; 5Division of Nephrology, Department of Internal Medicine, Keimyung University Dongsang Hospital, Keimyung University Kidney Institute, Daegu 42601, Korea; mdjin922@gmail.com

**Keywords:** protein-bound uremic toxin, indoxyl sulfate, p-cresyl sulfate, medium-cutoff, hemodiafiltration

## Abstract

The accumulation of protein-bound uremic toxins (PBUT) is associated with increased cardiovascular outcomes in patients on dialysis. However, the efficacy of PBUT removal for a medium-cutoff (MCO) membrane has not been clarified. This study was designed to assess the efficacy of PBUT clearance according to dialysis modalities. In this prospective and cross-over study, we enrolled 22 patients who received maintenance hemodiafiltration (HDF) thrice weekly from three dialysis centers. The dialysis removal of uremic toxins, including urea, beta 2-microglobulin (B2MG), lambda free light chain (λ-FLC), indoxyl sulfate (IS), and p-cresyl sulfate (pCS), was measured in the 22 patients on high-flux HD (HF-HD), post-dilution online HDF (post-OL-HDF), and MCO-HD over 3 weeks. The average convection volume in post-OL-HDF was 21.4 ± 1.8 L per session. The reduction rate (RR) of B2MG was higher in post-OL-HDF than in MCO-HD and HF-HD. The RR of λ-FLC was the highest in MCO-HD, followed by post-OL-HDF and HF-HD. The dialysate albumin was highest in MCO-HD, followed by post-OL-HDF and HF-HD. Post-dialysis plasma levels of IS and pCS were not statistically different across dialysis modalities. The total solute removal and dialytic clearance of IS and pCS were not significantly different. The clearance of IS and pCS did not differ between the HF-HD, post-OL-HDF, and MCO-HD groups.

## 1. Introduction

Gut microbiota are altered into toxic and less beneficial bacteria in uremic conditions [1,2]. Dominant gut microbial families in patients with chronic kidney disease (CKD) have shown an increase in the genes that generate p-cresols and indoles and a reduction in the genes that produce short-chain fatty acids, which are the main energy source of healthy colonocytes [3]. Thus, the concentration of gut-derived protein-bound uremic toxins (PBUTs) is increased in the circulation of patients with CKD [4,5]. Indoxyl sulfate (IS) and p-cresyl sulfate (pCS) are the most widely known PBUTs closely associated with increased cardiovascular disease in CKD [6,7]. Dietary tryptophan metabolizes to indole via intestinal bacteria and finally converts to IS in the liver, and tyrosine changes to pCS via microbial metabolism [8,9]. Even though PBUTs are small (<500 Da), they tend to bind to larger-sized proteins, particularly albumin, and the bound form is difficult to remove by diffusive dialysis [10]. Among PBUTs, IS and pCS are classified as PBUTs that are strongly and tightly bound with albumin (>90%) [11,12]. The plasma levels of PBUTs increases with CKD progression, whereas their fecal levels do not change [13]. Therefore, impaired renal function is suggested to be the main contributor to PBUT accumulation. The native kidney primarily removes PBUT by tubular secretion via organic anion or cation transporters [14,15]. Conventional hemodialysis (HD) partially replaces glomerular filtration but is unable to reproduce tubular function; thus, PBUTs build up in the plasma of patients on HD despite maintaining dialysis. However, the free forms of PBUTs, which occupy a very small fraction, are easily eliminated by diffusive HD [16].

Various HD strategies have been applied to efficiently remove PBUT. Prolongation of HD duration using nocturnal dialysis eliminated PBUTs to a greater extent [17]. Adding a convective method with hemodiafiltration (HDF) enhanced the removal of middle molecular uremic toxins, which led to a decrease in the cardiovascular mortality of patients on HD with a high convection volume [18]. However, better clearance of PBUTs was not consistent in high-volume HDF [19,20]. A recently developed medium cut-off (MCO) dialyzer selectively removes larger-sized uremic toxins. The performance of MCO-HD in removing middle- to large-sized toxins was comparable or occasionally superior to that of HDF. However, the efficacy of PBUT elimination in MCO-HD remains unclear. Therefore, this study was performed to test the effectiveness of dialysis for removing PBUTs in high-flux HD (HF-HD), high-volume post-dilution online HDF (post-OL-HDF), and MCO-HD.

## 2. Results

### 2.1. Patients and Dialysis Characteristics

The mean age of the patients was 62.18 ± 11.42 (Table 1). The incidence rate of diabetic end-stage renal disease was 45.5%, and the average blood flow was 302.73 ± 6.92 mL/min. Blood pressure and body weight were similar across the three dialysis methods; there were no differences in the dialysis parameters among the three dialysis modalities. The patients’ mean blood and dialysate flow rates were approximately 300 and 530 mL/min, respectively. The mean convection volume was 21.50 ± 1.90 L in post-OL-HDF. The mean ultrafiltration volume was 1.8–2.0 L/session. The characteristics of dialyzers are described in Table 2.

### 2.2. The Clearance of Small and Middle Molecular Weight Toxins

As for small molecular toxins, blood urea nitrogen and creatinine were significantly reduced after dialysis but there was no difference in their post-dialysis levels across the three different dialysis methods (Figure 1). The urea reduction rate (URR) (HF-HD: 78.36 ± 5.33%, Post-OL HDF: 80.32 ± 4.39%, and MCO-HD: 79.45 ± 4.86%; *p* = 0.417) and spKt/V (HF-HD: 1.86 ± 0.37, Post-OL HDF: 1.95 ± 0.33, and MCO-HD: 1.90 ± 0.32; *p* = 0.699) were also similar among the three dialysis modalities (Figure 2). For middle molecular weight toxins, B2MG (11.8 kDa) and λ-FLC (45 kDa) were selected. The post-dialysis B2MG level was lowest in post-OL-HDF (4.32 ± 1.21 mg/L, *p* < 0.001), followed by MCO-HD (5.27 ± 1.45 mg/L) and HF-HD (6.27 ± 1.70 mg/L). The RR of B2MG was higher in post-OL-HDF (79.54 ± 4.72%; *p* < 0.001) than in HF-HD (72.87 ± 3.98%) and MCO-HD (75.32 ± 4.64%).

However, there was no statistical difference in the RR of B2MG between the HF-HD and MCO-HD groups. The plasma concentration of λ-FLC was significantly reduced after dialysis in post-OL-HDF and MCO-HD, whereas no reduction was noted in HF-HD (Figure 1). The post-dialysis λ-FLC level was lower in post-OL-HDF (89.23 ± 34.09 mg/L, *p* < 0.001) and MCO-HD (71.59 ± 29.61 mg/L) than in HF-HD (137.73 ± 60.71 mg/L). The RR of λ-FLC was highest in MCO-HD (51.52 ± 6.08%; *p* < 0.001), followed by post-OL-HDF (43.48 ± 7.41%) and HF-HD (20.80 ± 8.14%) (Figure 2). The B2MG was significantly decreased after dialysis in all types of dialysis, whereas λ-FLC was not substantially eliminated in HF-HD (Figure 1).

### 2.3. Dialysate Albumin Removal

Pre-dialysis (HF-HD: 3.95 ± 0.21 g/dL, post-OL-HDF: 4.09 ± 0.29 g/dL, and MCO-HD: 3.91 ± 0.29 g/dL), post-dialysis plasma albumin (HF-HD: 4.18 ± 0.50 g/dL, post-OL-HDF: 4.18 ± 0.50 g/dL, and MCO-HD: 4.09 ± 0.43 g/dL), and RRs of albumin (HF-HD: −9.77 ± 11.09%, post-OL-HDF: −5.26 ± 7.73%, and MCO-HD: −5.50 ± 9.17%) were not significantly different among the three different dialysis modalities. However, the dialysate albumin mass was highest in MCO-HD (2547.32 ± 968.31 mg/session, *p* < 0.001), followed by post-OL-HDF (778.32 ± 313.17 mg/session) and HF-HD (59.91 ± 70.82 mg/session) (Figure 2).

### 2.4. The Clearance of Protein-Bound Uremic Toxins

The post-dialysis IS and pCS were significantly reduced compared to those in pre-dialysis in the three different dialysis modalities (Figure 1). However, the plasma concentrations of IS and pCS were similar in the three modalities in pre-dialysis (IS, HF-HD: 22.59 ± 10.41 mg/dL, post-OL-HDF: 21.39 ± 11.05 mg/dL, and MCO-HD: 20.08 ± 9.46 mg/dL; pCS, HF-HD: 37.79 ± 18.45 mg/dL, post-OL-HDF: 38.70 ± 22.65 mg/dL, and MCO-HD: 33.09 ± 17.47 mg/dL) and post-dialysis (IS, HF-HD: 15.27 ± 7.60 mg/dL, post-OL-HDF: 12.90 ± 6.83 mg/dL, and MCO-HD: 13.12 ± 7.12 mg/dL; pCS, HF-HD: 26.60 ± 12.32 mg/dL, post-OL-HDF: 24.80 ± 14.08 mg/dL, and MCO-HD: 23.39 ± 13.68 mg/dL). In addition, there was no statistical difference in the RR, dialysate mass removal, and dialytic clearance of IS and pCS among the three different dialysis methods (Figure 3). In HF-HD, post-OL-HDF, and MCO-HD, the RR of IS was 33.50 ± 11.48%, 40.03 ± 10.16%, and 36.31 ± 12.75%; the dialysate mass of IS was 94.98 ± 96.01 mg, 83.63 ± 100.50 mg, and 74.31 ± 66.79 mg; and the IS-clearance was 21.12 ± 19.91 mL/min, 19.50 ± 17.22 mL/min, and 19.49 ± 15.81 mL/min, respectively. The RR of pCS was 27.06 ± 10.74%, 34.44 ± 10.00%, and 29.49 ± 10.28%; the dialysate mass of pCS was 114.56 ± 12.39 mg, 126.19 ± 69.54 mg, and 101.71 ± 57.43 mg; and the pCS-clearance was 13.58 ± 2.76 mL/min, 15.33 ± 3.17 mL/min, and 14.10 ± 3.94 mL/min, respectively. There was no correlation between the dialysate mass of IS or pCS and dialysate albumin concentration (Figure 4).

## 3. Discussion

This study demonstrated that highly efficient dialysis, including MCO-HD and post-OL-HDF, did not show better performance in eliminating IS and pCS compared to that of HF-HD, despite higher albumin removal during a dialysis session. PBUTs mainly exist in circulation in the protein-bound form where albumin is the primary protein. The albumin-bound capacity varies depending on the characteristics of the PBUTs. IS and pCS are known to be highly bound to albumin, accounting for more than 90%, and their free forms account for approximately 2% in the circulation of normal individuals [16]. HD cannot replace renal tubular secretion, which is the main mechanism to remove PBUT, and thus PBUTs cannot be easily removed by HD. Consequently, the plasma levels of IS and pCS are 50–100 times higher in patients on dialysis than in those with normal kidney function [16]. High-efficiency HD, such as MCO-HD and high-volume HDF, was supposed to have better performance in clearing protein-bound toxins by greater removal of albumin during a dialysis session. 

However, three randomized controlled cross-over studies have reported inconsistent results. One study showed that the plasma concentration of total pCS was significantly lower and its dialysis mass removal and dialytic clearance were higher in pre-dilution HDF with 60 L convection volume than in HF-HD, while post-dilution HDF with 20 L convection volume failed to achieve a better function [20]. In addition to PBUTs, small molecular weight uremic toxins, including urea nitrogen and creatinine, were eliminated better in pre-dilution HDF (60 L) than in post-dilution HDF (20 L). The enhanced capacity to remove free-form PBUTs and small toxins in the pre-dilution HDF with 60 L might be the reason for the augmentation of PBUTs’ removal. Another study reported a slightly elevated RR of free and total forms of pCS and IS in post-dilution HDF (24.3 L compared to those in HF-HD) [19]. Nevertheless, there was no difference in the pre-dialysis plasma concentrations of pCS and IS. A previous study demonstrated that the RR of total pCS and IS significantly increased, but the plasma concentrations did not change when the dialysis duration was extended from 4 to 8 h [17]. The results were independent of the addition of convective dialysis. A non-randomized study suggested better function in the clearance of PBUTs, showing reduced plasma levels of pCS in patients on dialysis when they changed their dialysis from HF-HD to post-dilution HDF with a 19 L substitutional volume [21]. However, the data may be unreliable because the results were not validated via a cross-over trial. In our study, we found that the plasma level and clearance function of IS and pCS did not change even after the addition of sufficient convective dialysis or the use of the MCO dialyzer.

The native kidney removes PBUTs mainly as free forms, and the clearance of total forms of IS and pCS are only 2.0 and 1.7%, respectively [16]. Dialytic clearance of the total forms of IS and pCS was slightly lower or similar to that of the native kidney, while dialytic clearance of the free forms was only 20–30% of that of the native kidney [16]. In contrast, the dialytic clearance of urea was 4–5 times higher than that of the native kidney. Despite the higher clearance, the plasma urea of patients on dialysis was on average four times higher than that of healthy controls due to the dialysis time limit (usually 4 h three times a week). Unlike urea, the dialytic clearance of IS and pCS is only one-fifth of the native kidney clearance; thus, the plasma levels of IS and pCS were 116 and 41 times higher, respectively, than those of healthy participants [16]. The clearance of the total form of IS has been reported to be similar between HD and the native kidney. Therefore, greater albumin removal via the MCO dialyzer or the convective method might be futile to remove higher total forms of IS. For pCS, dialytic clearance of the total form was reported to be 55% of that of the native kidney clearance. If we could accomplish greater removal of albumin via dialysis, 100% of the total pCS would be removed. However, the effect of the removal of total pCS on the change in plasma concentration of pCS was very small. The PBUT clearance increased when the higher free form of PBUT was removed in the diffusive dialysis with prolonged dialysis time, whereas it did not change even when convective dialysis was added [20]. The study did not show a decrease in the plasma concentrations of PBUTs, despite augmentation of clearance. Since dissociation from the bound form to the free form requires sufficient time, a dialysis session might be too short to restore the new equilibrium for strongly bound PBUTs, such as IS and pCS [11]. In addition, the concentrations of albumin and PBUTs in the spent dialysate did not correlate with HDF [20,22]. Similarly, we confirmed no correlation between dialysate albumin and IS or pCS (Figure 4). This means that the removed albumin in HDF might not be helpful in eliminating PBUTs in HDF. 

Researchers have attempted other methods for removing PBUTs. Oral activated carbon, AST-120, removes PBUTs via absorption of intestinal precursors and has shown beneficial effects in reducing cardiovascular events and mortality in patients with CKD [23]. The amino acids derived from animal proteins contribute to alterations of the gut microbiota and excessive generation of uremic toxins [24]. Therefore, a plant-based diet was suggested to lead to beneficial changes in the gut microbiota. One study showed that a high-fiber and low-protein diet was associated with low levels of PBUTs in CKD patients [25]. However, the maintenance of a high-fiber diet is challenging for anuric ESRD patients due to occurring hyperkalemia. Supplementation of probiotics showed beneficial effects in reducing plasma IS and indoxyl glucuronide, while it was not effective for reducing pCS levels [26,27]. The infusion of ibuprofen, as a competitive binder for PBUTs, during the HD session significantly augmented the dialytic clearance of PBUTs and mitigated the plasma levels [28].

The clearance of large uremic toxins was better in MCO-HD than that in post-OL-HDF. MCO-HD had a higher λ-FLC clearance than that of post-OL-HDF. However, the clearance of B2MG was superior in post-OL-HDF than that in MCO-HD. In line with this, other studies demonstrated better removal of λ-FLC and less clearance of B2MG in MCO-HD compared with that in HDF [29,30]. The dialyzer characteristics of the MCO membrane, which is efficient in removing medium to large molecular weight solutes, might explain the different clearing efficacy depending on the molecular size [31]. Expanded HD refractorily loses albumin instead of higher uremic toxin removal [31]. However, it is unclear whether albumin removal is greater in MCO-HD or HDF. This might differ depending on the dialyzers used in HDF [29,32]. We observed that dialysate albumin was the highest in MCO-HD compared with those of post-OL-HDF and HF-HD. Some toxins, such as α1-microglobulin, have a similar removal rate to albumin clearance [33]. Thus, several researchers have suggested that albumin leakage over 3 g per dialysis session might be beneficial for removing more toxins as a trade-off [34]. However, even with a larger loss of albumin, MCO-HD did not show beneficial results for the clearance of IS and pCS in this study. In contrast to our expectation that the MCO dialyzer might be better at removing PBUTs, this study first clarified that there was no significant difference in eliminating PBUTs, even with the MCO membrane.

This study has some limitations. First, we did not measure the free forms of IS and pCS. Since the solute kinetics of bound and free PBUTs are different, we might obtain more precise information regarding dialytic clearance if the free forms are measured. However, dialytic clearance was mainly accomplished with free PBUTs and their blood level was low, thus the effect of their changes compared with total ones might be subtle. In addition, the clearance of the free forms of IS and pCS might be similar since their removal pattern was similar to those of total forms even after dialysis methods (HD vs. HDF) or time (4 vs. 8 h) were changed [17,19]. Second, we collected the dialysate four times every hour. Albumin elimination is higher in the first 90 min of the HD session [35]. Considering albumin kinetics, the real albumin loss via dialysis might be larger than our measurement. Nevertheless, this study is meaningful because it is the first trial to compare PBUTs’ removal across three different dialysis methods, including MCO-HD, using a randomized cross-over design study.

## 4. Conclusions

We could not find any superiority of post-OL-HDF or MCO-HD in clearing IS and pCS.

## 5. Materials and Methods

### 5.1. Study Designs and Patient Selection

In this prospective, randomized, cross-over study, 22 patients who received maintenance HDF thrice weekly for a period of >3 months were enrolled from three tertiary dialysis centers (Kyung Hee University Hospital at Gangdong, Konkuk University Hospital, and Keimyung University Dongsan Hospital) from June to July 2021 (CRIS No. KCT0007587). Adult patients aged 18 years or older who were anuric and had permanent vascular access were included in the study. Patients who had residual urine of more than 100 cc/day, were pregnant, received dialysis with a catheter, had malignancy, congestive heart failure, or hemodynamic instability were excluded. Anthropometric and dialysis-related information and laboratory findings were collected during study entry. This study was approved by the Institutional Review Board (KHNMC 2021-03-063-004), and informed consent was obtained from all patients.

### 5.2. Treatment Protocols

Seven to eight patients were randomly subjected to thrice-weekly HF-HD, post-OL-HDF, and MCO HD, for three consecutive weeks each. Plasma and dialysate samples were collected during the mid-week treatment in the third week. Fx CorDiax 80 (Fresenius Medical Care Deutschland, Bad Homburg, Germany), Theranova 400 (Gambro Dialysatoren GmbH, Hechingen, Germany), and Fx CorDiax 800 (Fresenius Medical Care Deutschland, Bad Homburg, Germany) dialyzers were used for HF-HD, MCO-HD, and post-OL-HDF. The dialysis duration was 4 h, with a dialysate flow of 500 mL/min and a blood flow of 250–320 mL/min. HDF was performed in post-dilution mode with a target convective volume of ≥23 L.

### 5.3. Measurement Methods

Blood samples were taken before the dialysis onset and immediately after the dialysis session in a mid-week treatment in the 3rd week. Dialysate mixtures were collected from the inverse pump at 60, 120, 180, and 240 min of the dialysis session. A total of 10 mL of the dialysate sample from the mixture was obtained. All samples were stored at −80 °C until further use. The plasma levels of urea nitrogen (blood urea nitrogen, 60 Da), λ-free light chain (λ-FLC, 45,000 Da), and β2-microglobulin (B2MG, 11,800 Da) were measured before and after dialysis. A chemiluminescence immunoassay was used to measure B2MG (Immunlites 2000 XPI, Seimenes Healthcare Diagnostics Products, Gwynedd, UK), and a turbidimetric immunoassay was used to measure λ-FLC (SPA PLUS, The Binding Site, Burmingham, UK). The plasma albumin and protein levels were monitored before and after dialysis. The dialysate albumin level was measured via the enzyme-linked immunosorbent assay (Alpha Diagnostic, San Antonio, TX, USA). Simultaneous quantitative analyses of the total IS (212 Da, protein-bound ~90–95%) and pCS (187 Da, protein-bound ~95%) in the plasma and dialysate were performed with slight modifications using liquid chromatography-tandem mass spectrometry (LC-MS/MS) [36,37,38,39]. Briefly, 25 μL of human plasma or 125 μL of dialysate was precipitated with acetonitrile, including internal standards (pCS-d7), and vortexed for 30 s, followed by centrifugation for 10 min at 19,500× *g* at 4 °C. Next, the supernatant (200 μL plasma sample and 300 μL dialysate sample) from each tube was transferred to a new microcentrifuge tube and evaporated using a Speed-Vac (HyperVac, VC2200, Gyrozen Co., LTD, Deajeon, Korea). Subsequently, 50 μL and 25 μL of solvent A (5 mmol/L ammonium acetate solution) were used for the dried plasma and dialysate samples, respectively. For quantification of target uremic toxins, LC-MS analysis was performed using a triple quadrupole mass spectrometer (6490 series, Agilent Technologies, Wilmington, DE, USA) coupled to a 1200 series high-performance liquid chromatography system (Agilent Technologies, Wilmington, DE, USA) with a Hypersil GOLD column (2.1 × 100 mm ID; 1.9 μm, Thermo Fisher Scientific, Waltham, MA, USA). The injection volume was 2 μL for serum and 3 μL for dialysate analyses. The total run time was 14.5 min for each analysis and the LC gradient system was performed as follows: 0–1 min, 20% solvent B (100% methanol); 1–2.5 min, 20–60% solvent B; 2.5–3 min, 60–95% solvent B; 3–5 min, 95% solvent B; 5–5.5 min, 95−20% solvent B; and 5.5–14.5 min, 20% solvent B with a fixed LC flow rate of 0.15 mL/min. IS and pCS were eluted at 2.8 min and 4.3 min, respectively. The electrospray (ESI) MS method was used to analyze IS and pCS, and all acquisition method parameters were set as follows: capillary voltage: 3000 V in negative mode, drying gas flow: 12 L/min at 290 °C, sheath gas flow: 12 L/min at 400 °C, and nebulizer gas flow at 30 psi. Multiple reaction monitoring (MRM) conditions, including MS/MS collision energy and computed transitions, were optimized for each molecule to analyze the target uremic toxins in individual samples. The MRM transition was selected at the following transitions: *m*/*z* 212.04→80.14, 132.05 for IS, *m*/*z* 186.98→80.02, 107.03 for pCS, and *m*/*z* 194.04→80.02, 114.04 for pCS-d7. The charcoal-stripped human plasma and dialysate before dialysis served as blank matrices for constructing the calibration curves. All experiments were performed in triplicate, and the raw data were processed using the Skyline software package (version 21.2.0.369, MacCoss Lab, University of Washington, Seattle, WA, USA) and the Mass Hunter Workstation Data Acquisition software (Agilent Technologies, Wilmington, DE, USA).

### 5.4. Calculation

The reduction rate (RR) of the solutes was defined as (RR = [1 − (C_post_/C_pre_)] × 100, C_post_ = post-dialysis plasma concentration, C_pre_ = pre-dialysis plasma concentration). C_post_ was calculated using a reference formula reflecting hemoconcentration [40]. For the middle-molecular-weight toxins, C_post_ was replaced by C_post-corr_ (C_post-corr_ = C_post_/ [1 + (BW_pre_ − BW_post_)/(0.2 × BW_post_)], BW_pre_ = body weight before dialysis, BW_post_ = body weight after dialysis). Total solute removal (TSR) was calculated by multiplying the solute concentrations in the 10 mL dialysate by the effluent volume (dialysate, ultrafiltration, and substitution volume). Dialytic clearance was attained by dividing the TSR by dialysis duration. Single-pool Kt/V(spKt/V) was assessed using the reference method [41].

### 5.5. Statistics

All data are presented as the mean values ± standard deviations. Comparisons of the three groups were performed using analysis of variance (ANOVA), and a significant difference was determined using Fisher’s least significant difference test. Statistical analyses were conducted using SPSS software (version 20, SPSS, Inc., Chicago, IL, USA). Statistical significance was set at *p* < 0.05.

## Figures and Tables

**Figure 1 toxins-14-00779-f001:**
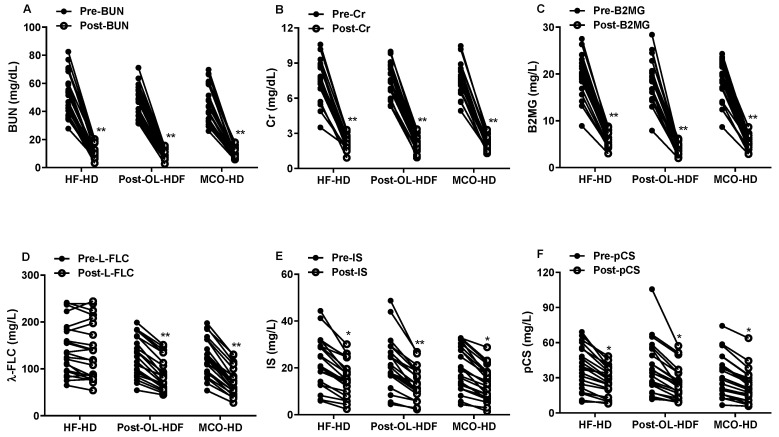
The plasma concentrations of uremic toxins before and after dialysis. (**A**): Blood urea nitrogen (BUN), (**B**): Creatinine (Cr), (**C**): λ-free light chain (λ-FLC), (**D**): β2-microglobulin (B2MG), (**E**): Indoxyl sulfate (IS), and (**F**): p-cresyl sulfate (pCS) plasma levels before and after dialysis in high-flux hemodialysis (HF-HD), post-dilution online hemodiafiltration (post-OL-HDF), and medium cut-off hemodialysis (MCO-HD). * *p* < 0.05 vs. the level of pre-dialysis, ** *p* < 0.01 vs. the level of pre-dialysis.

**Figure 2 toxins-14-00779-f002:**
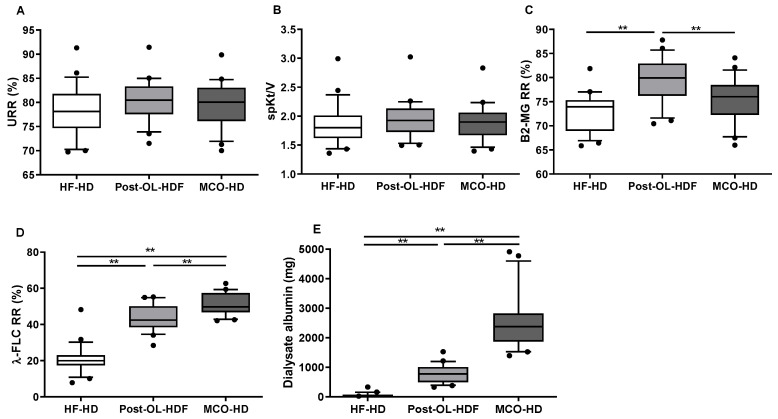
The reduction rates of uremic toxins and dialysis albumin. (**A**): Urea reduction rate (URR), (**B**): Single pool Kt/v (spKt/v), (**C**): Reduction rate (RR) of β2-microglobulin (B2MG), (**D**): RR of λ-free light chain (λ-FLC), (**E**): Dialysis albumin mass in high-flux hemodialysis (HF-HD), post-dilution online hemodiafiltration (post-OL-HDF), and medium cut-off hemodialysis (MCO-HD). ** *p* < 0.01 vs. other dialysis method.

**Figure 3 toxins-14-00779-f003:**
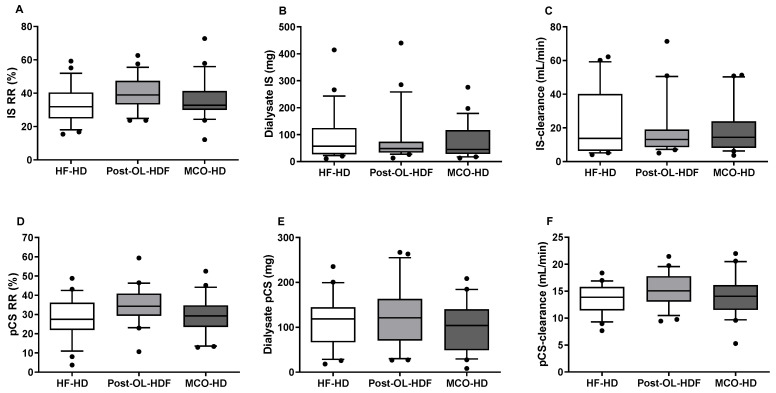
The clearing efficacy of IS and pCS. (**A**,**D**): Reduction rate (RR) of Indoxyl sulfate (IS) and p-cresyl sulfate (pCS); (**B**,**E**): Dialysate removed IS and pCS; (**C**,**F**): Dialytic clearance of IS and pCS mass in high-flux hemodialysis (HF-HD), post-dilution online hemodiafiltration (post-OL-HDF), and medium cut-off hemodialysis (MCO-HD).

**Figure 4 toxins-14-00779-f004:**
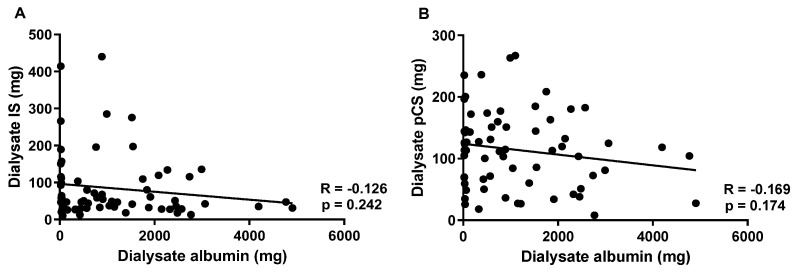
The association between dialysate albumin and dialysate PBUTs. (**A**): The correlation between dialysate albumin and dialysate IS, (**B**): the correlation between dialysate albumin and dialysate pCS in the pooled samples from the three different HD methods. R: Pearson correlation coefficient.

**Table 1 toxins-14-00779-t001:** Baseline characteristics of the patients.

Variable	Results
Age	62.2 ± 11.4
Sex (M:F)	11:11
BMI	23.0 ± 2.7
Dialysis vintage (months)	67.1 ± 41.3
Causative disease of ESRD (number,%)	
Diabetes	10, 45.5
Hypertension	9, 40.9
Glomerulonephritis	1, 4.5
Polycystic kidney	2, 9.1
Dry weight (kg)	59.5 ± 7.5
Blood flow (mL/min)	302.7 ± 6.9
Pre-dialysis SBP (mmHg)	138.7 ± 22.2
Pre-dialysis DBP (mmHg)	65.5 ± 8.5
Hb (mg/dL)	11.1 ± 1.2
Albumin (g/dL)	4.0 ± 0.2
BUN (mg/dL)	51.2 ± 15.2
Cr (mg/dL)	7.6 ± 2.0

BMI, body mass index; ESRD, end-stage renal disease; SBP, systolic blood pressure; DBP, diastolic blood pressure; and BUN, blood urea nitrogen.

**Table 2 toxins-14-00779-t002:** The characteristics of the dialyzers.

	HF-HD	Post-OL-HDF	MCO-HD
Dialyser	Fx CorDiax 80	Fx CorDiax 800	Theranova 400
Inner diameter (μm)	185	210	180
Wall thickness (μm)	35	35	35
Membrane polymer	polysulphone-PVP blend	polysulphone-PVP blend	polyarylethersulphone-PVP blend
Effective surface area (m^2^)	1.8	2.0	1.7
Sieving coefficients of albumin	<0.001	<0.001	0.008
Clearance of inulin	135	178	183
UF coefficient (mL/h/mmHg)	64	62	48

PVP, polyvinylpyrrolidone; UF, ultrafiltration.

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
