# Peer review of "The Medium Cut-Off Membrane Does Not Lower Protein-Bound Uremic Toxins"

_toxins, 2022, doi:10.3390/toxins14110779_

Round 1

Reviewer 1 Report

In this work the authors compare the removal of urea, B2MG, λ-FLC, IS, and pCS from patients undergoing three different dialysis modalities: high-flux HD(HF-HD), post-dilution online HDF (post-OL-HDF), and medium cutoff (MCO-HD). For HF-HD the hemodialyzer used was the x CorDiax 80 (Fresenius Medical Care Deutschland, Bad Homburg, Germany); for post-OL-HDF the Theranova 400 (Gambro Dialysatoren GmbH, Hechingen, Germany) hemodialyzer was used, and for the MCO-HD, the Fx CorDiax 800 (Fresenius Medical Care 252 Deutschland, Bad Homburg, Germany) hemodialyzer was used.

The manuscript is well-structured and relevant for the field. The introduction is clear, and in the discussion section there are references to three other randomized controlled crossover studies which had contradictory findings.

In general, the article is well organized, and the conclusions add important information to the results obtained in the past. A more comprehensive overview of the properties of each of the hemodialyzers (KUF, MWCO, membrane material, etc.) used in this study would add quality to the manuscript as opposed to only referencing them in section 5.2 of the Materials and Methods section.

Other minor comments which may help improve the already excellent quality of the manuscript:

Page 6 Lines 171-174

If the dialytic clearance of urea was 4–5 times higher than that of the native kidney why is the plasma urea of HD patients 4x higher than that of healthy patients?

Author Response

1.

In general, the article is well organized, and the conclusions add important information to the results obtained in the past. A more comprehensive overview of the properties of each of the hemodialyzers (KUF, MWCO, membrane material, etc.) used in this study would add quality to the manuscript as opposed to only referencing them in section 5.2 of the Materials and Methods section.

: I appreciate your valuable comments. We add table 2 with comments, which contains detailed information regarding hemodialyzers in the result part.

The MVCO value of Theranova 400 was 56kDa. However, the manufacturer (FMC) does not provide MWCO values in Fx CorDiax membranes. Therefore, we do not include MVCO values in this table.

Table 2. The characteristics of dialyzers

HF-HD

Post-OL-HDF

MCO-HD

Dialyser

Fx CorDiax 80

Fx CorDiax 800

Theranova 400

Inner diameter (μm)

185

199

180

Wall thickness (μm)

38

44

35

Membrane polymer

polysulphone-PVP blend

polysulphone-PVP blend

polyarylethersulphone-PVP blend

Effective surface area (m2)

1.8

2

2

Sieving coefficients of albumin

< 0.001

< 0.001

0.008

Clearance of inulin (QB =300mL/min)

135

156

161

UF coefficient (mL/h/mmHg)

64

64

59

2.

Other minor comments which may help improve the already excellent quality of the manuscript:

Page 6 Lines 171-174

If the dialytic clearance of urea was 4–5 times higher than that of the native kidney why is the plasma urea of HD patients 4x higher than that of healthy patients?

: I appreciate your comments.

Due to the dialysis time limit (usually HD patients have only 4 hours of dialysis three times a week), plasma urea concentration usually is higher in dialysis patients than in healthy controls despite the higher clearance of urea. We insert this comment in the discussion part.

Before) The plasma urea of patients on dialysis was on average four times higher than that of healthy controls because urea was eliminated via intermittent HD.

After) Despite the higher clearance, the plasma urea of patients on dialysis was on average four times higher than that of healthy controls due to the dialysis time limit (usually 4 hours three times a week). 

Reviewer 2 Report

This article mentioned the removal of toxins in vivo by MCO membrane compared with conventional dialysis and Post Online HDF. 

I think the novel finding in this study is the investigation of  MCO membrane. So it is necessary to describe the characteristics of this membrane in more detail from an engineering point of view, such as  UFR, removal characteristics of several representative indicator substances,  sieving coefficient of albumin which described in the product specifications. The presentation of other two membrane specification are also helpful to understand the results described subsequently.

 Authors well explained uremic toxin in introduction. However, the description was unclear that the common mechanism and different mechanism to remove protein binding uremic toxins among three methods.

Results:  Appropriately described.

Discussion: The study limitation that free form of toxins were not measured is certainly a critical concern. In summary, the removal of free form of uremic toxins mainly performed by diffusion, it is not strange that no significant difference among dialysis mode in this study. There should be more discussion about the significance of this research.

Author Response

1.

I think the novel finding in this study is the investigation of  MCO membrane. So it is necessary to describe the characteristics of this membrane in more detail from an engineering point of view, such as  UFR, removal characteristics of several representative indicator substances,  sieving coefficient of albumin which described in the product specifications. The presentation of other two membrane specification are also helpful to understand the results described subsequently.

: I appreciate your valuable comments. We add table 2 with comments, which contains detailed information regarding hemodialyzers in the result part.

Table 2. The characteristics of dialyzers

HF-HD

Post-OL-HDF

MCO-HD

Dialyser

Fx CorDiax 80

Fx CorDiax 800

Theranova 400

Inner diameter (μm)

185

199

180

Wall thickness (μm)

38

44

35

Membrane polymer

polysulphone-PVP blend

polysulphone-PVP blend

polyarylethersulphone-PVP blend

Effective surface area (m2)

1.8

2

2

Sieving coefficients of albumin

< 0.001

< 0.001

0.008

Clearance of inulin (QB =300mL/min)

135

156

161

UF coefficient (mL/h/mmHg)

64

64

59

2.

Authors well explained uremic toxin in introduction. However, the description was unclear that the common mechanism and different mechanism to remove protein binding uremic toxins among three methods.

: I appreciated your important comment.

The total forms of PBUTs were not easily removed by conventional HD methods since they could be eliminated by tubular secretion via organic transporters. However, the free forms of PBUTs were removed well only through the diffusive method of HD. We already described the detailed mechanisms in the discussion part. We edited some ambiguous terminology in the introduction part.  

Before) thus, PBUTs build up in the plasma of patients on HD despite maintaining dialysis. However, their free forms, which occupy a very small fraction, are easily eliminated by HD.

After) thus, PBUTs build up in the plasma of patients on HD despite maintaining dialysis. However, the free forms of PBUTs, which occupy a very small fraction, are easily eliminated by diffusive HD.

Results:  Appropriately described.

3.

Discussion: The study limitation that free form of toxins were not measured is certainly a critical concern. In summary, the removal of free form of uremic toxins mainly performed by diffusion, it is not strange that no significant difference among dialysis mode in this study. There should be more discussion about the significance of this research.

: I appreciated your valuable commnets.

We inserted some comments and edited several sentences in the discussion part.

Insertion) Opposing our expectation that the MCO dialyzer might be a better function to remove PBUTs, this study firstly clarified that there was no significant difference in eliminating PBUTs even with the the MCO membrane. We insert this comment in the discussion part.

Before) However, we can suppose that the clearance of the free forms of IS and pCS might be similar because their removal is mainly performed in free forms.

After) However, dialytic clearance was mainly accomplished with free PBUTs and their blood level was low, thus the effect of their changes compared with the total ones might be subtle. In addition, the clearance of the free forms of IS and pCS might be similar since their removal was a comparable pattern to those of total forms even after dialysis methods (HD vs. HDF) or dialysis time (4 vs. 8 hours) were changed.

Reviewer 3 Report

The manuscript assess the efficacy of protein-bound uremic toxins clearance according to dialysis modalities. The sample size , parameters anaysed are appropriate and well supported with relevant statistical analysis and authors also report the limitations of their study. I feel they have done good work and I recommend acceptance in the current form.

Author Response

I appreciate your valuable comments. 

Round 2

Reviewer 2 Report

Thank you for your revision. I recommend that it be accepted for publication.